# The Great Wall Vibration Monitoring of Traffic and Natural Hazards Using MEMS Accelerometers

**DOI:** 10.3390/s23042179

**Published:** 2023-02-15

**Authors:** Jian Wang, Xu Liu, Fei Liu, Cai Chen, Yuyang Tang

**Affiliations:** 1School of Geomatics and Urban Spatial Informatics, Beijing University of Civil Engineering and Architecture, Beijing 102616, China; 2Research Center for Urban Big Data Applications, Beijing University of Civil Engineering and Architecture, Beijing 100044, China

**Keywords:** MEMS accelerometer, traffic vibration, thunder loading, vibration monitoring

## Abstract

The stability of the Great Wall is mainly affected by traffic vibrations and natural hazards, such as strong winds, heavy rainfall, and thunderstorms, which are extremely harmful to the safety of the Great Wall. To determine the impact of the above factors on the Great Wall, a comparative analysis based on MEMS (micro-electro-mechanical system) accelerometer data was conducted between the non-impacts and the impacts of the above factors. An analysis of the relationship between vibration acceleration and each potential hazard based on a visual time series chart was presented using the data of accelerometers, traffic video, meteorology, rainfall, and wind. According to the results, traffic vibration is one of the primary dangerous factors affecting the stability of the Great Wall, Moreover, the intensity of the vibrations increases with the traffic flow. Thunderstorms also influence the stability of the Great Wall, with enhanced thunderstorm excitation resulting in increased vibration displacement. Furthermore, wind load is an influencing factor, with average wind speeds greater than 9 m/s significantly affecting the stability of the Great Wall. Rainfall has no impact on the stability of the Great Wall in the short term. This research can provide important guidance for risk assessment and protection of the Great Wall.

## 1. Introduction

The Great Wall of China, one of the architectural marvels, natural landscapes, and cultural heritage sites in human history, has outstanding universal value in the history of human civilization and was listed as a world cultural heritage in 1987 [1]. With the rapid development of transportation systems, the number of highways crossing the Great Wall is increasing [2]. For example, the Erdaoguan Great Wall in Huairou District is intersected by Ansi Road, the only highway with a large traffic flow for people traveling from Huairou to Yanqing in Beijing. Although the vibrations induced by traffic are small, the old Great Wall has grown weaker in seismic resistance due to accumulated damage. Therefore, it is more likely to be deformed than other buildings under the influence of long-term continuous vibrations [3,4].

Many scholars in the field of protection of ancient buildings have carried out extensive research on how traffic vibrations can affect the Great Wall and it has been shown, through the use of finite element analysis modelling, that the effects can be larger if the Great Wall is near the road [5,6,7,8,9]. However, it is difficult to practically obtain accurate structural parameters of the Great Wall, such as masonry materials, weight, and building height, which are essential for finite element analysis methods [10,11]. Moreover, it is only theoretically possible to analyze the results. Chen et al. used a seismograph as the main instrument used in field tests to measure the properties of ground vibrations. It was concluded that the vibration of the Great Wall caused by ground traffic increases with the increasing speed and weight of the vehicle and decreases with the increasing distance of the vibration source [12]. However, this result does not take into account the effects of weather such as wind and rain.

In addition to traffic vibration, the stability of the Great Wall is affected by natural factors such as strong wind, heavy rain, and thunderstorms [13,14,15,16]. Gao et al. conducted a shaking table test to simulate the impact of earthquakes on the Xi’an Ancient City Wall and its seismic performance [17]. Xu et al. investigated the impact of blasting vibration in tunnel excavation on the Ming Great Wall above the tunnel using the finite element analysis method [18]. Wu et al. established a complete set of shadow classification systems for the deformation of ancient city walls first and summarized 11 typical influencing factors such as heavy rain erosion, wind erosion and spalling, and temperature using numerical simulation analysis [19], while ignoring field tests.

With the development of micro-electromechanical systems (MEMS), MEMS accelerometers with small volume, low cost and low power consumption are gradually replacing large, complex, and expensive seismometers [20,21,22,23]. The analysis of wind-induced vibrations on HVTL conductors by Zanelli et al. using a MEMS accelerometer showed that different wind-induced phenomena can be detected by the MEMS accelerometer [14]. Ferguson et al. proposed a vehicle detection method based on the generalised variance computed over a sliding window using a triaxial MEMS accelerometer [24]. However, it suffers from a deficiency in the high-frequency noise component, which was subsequently found to be insufficient for our purposes. Patanè et al. investigated low-noise and ultra-low-noise MEMS accelerometers with the ultra-low noise M−A352 Q−MEMS accelerometer in situ on the ground floor of a historic and strategic building in the city center. A series of tests have in fact shown that such a cost-effective accelerometer could play a key role in monitoring low-level seismicity and could be used in urban observatories and for SHM [25]. In this paper, smaller 3-axis low-cost and low-noise MEMS accelerometers were used to obtain the vibration data of the Great Wall. A comprehensive analysis based on a visual time series chart was carried out to explore the impact of traffic vibrations, wind loads, and rain loads on the Great Wall. The research results could underpin security monitoring and protection of the Great wall from disasters.

## 2. Equipment for Vibration Monitoring

In September 2021, a series of GNSS/accelerometer vibration monitoring systems were installed to collect GNSS (1 Hz) and accelerometer data (100 Hz) in the Great Wall of three sections. In total, three monitoring systems were installed at Erdaoguan Great Wall (Stations 63, 64, and 65), a monitoring system was installed at Qilianguan Great Wall (Station 67), and a monitoring system was installed at Hefangkou Great Wall (Station 66) (Figure 1). This test was dedicated to the analysis of the accelerometer data, and only the acceleration data from Stations 65 and 67 were analyzed.

To explore the relationship between the stability of the Great Wall and the impacts of traffic vibration, wind load, thunder load, and rainfall load, respectively, SCA3300 series 3-axis MEMS accelerometers were distributed fixed on enemy platforms of the Erdaoguan Great Wall in Huairou District, Beijing in December 2021. The sampling frequency of MEMS accelerometer data was set at 100 Hz. The low-frequency noise component of the accelerometers is removed by a de-trend algorithm. The parameters of the MEMS accelerometers are shown in the following Table 1.

To analyze how traffic vibrations can influence on the Great Wall, two 3-axis MEMS accelerometers were installed on the enemy platform so that one was further from the road and the other was closer to the road (see Figure 2), with the latter able to completely record the traffic information across the Great Wall (see Figure 3). In addition, a video camera monitor was fixed on the near-road monitor point (Station 65) to collect information about vehicle flow.

## 3. Analysis of Potential Hazards

### 3.1. Analysis of Traffic Vibrations Impact

To evaluate the impact of traffic vibrations on the Great Wall, a comparative analysis of the accelerometer data collected by the near-road Station 65 and the far-road Station 67 was performed. To minimize the interference of winds, we collected data only when the average wind speed on 31 January 2022 was between 1 to 2 m/s (Website for weather history: China Meteorological Data Service Centre https://data.cma.cn, accessed on 1 January 2022). Figure 4 shows the acceleration time-history curves collected by the above two stations. The acceleration time-history curve of Station 67 is very smooth with no fluctuations, and its maximum vibration acceleration is only 0.004; while the acceleration of Station 65 exhibits sharp and continuous fluctuations with a maximum amplitude of about 0.06 m/s^2^.

The traffic flow information was collected on 31 January 2022 through a fixed video camera monitor placed along Ansi Road to verify the impact of traffic vibrations on the Great Wall. The experimenter recorded traffic flow (the number of cars per hour) by counting. Table 2 shows the hourly traffic from 6:00 a.m. to 10:00 p.m.

As suggested by Figure 5, the larger the amplitude of acceleration time-history curve, the more vehicles pass through the road. More specially, the highest fluctuation of acceleration time-history curve coincides with the maximum traffic flow between 1:00 p.m. to 2:00 p.m. Thus, a strong correlation exists between acceleration and traffic flow, and vehicle traffic has a large impact on the safety of the Great Wall by exciting it to vibrate.

### 3.2. Analysis of Wind Load Impact

To analyze the impact of wind load on the Great Wall, a comparative analysis of accelerometer data collected on windy and windless days separately was performed. We downloaded the speed data from the website of China Meteorological Data Service Centre (CMDSC: https://data.cma.cn, accessed on 1 January 2022). The data of 1 January 2022 shows that high winds blew through Beijing’s Huairou district at a speed ranging from 12 m/s to 13 m/s. On 5 January 2022, there was almost no wind, with an average wind speed of only 2 m/s. Figure 6 illustrates that acceleration time-history curve on windy days is non-stationary with large fluctuations, while those for windless days is stable with no fluctuation. As a result, there is a correlation between acceleration and wind load, which can partially affect the Great Wall.

Moreover, to further analyze the degree of correlation between acceleration and wind load, we downloaded the wind speed data per half hour from the website of Weather for 243 Countries of the World (https://rp5.ru/, accessed on 5 January 2022 and 10 January 2022). As seen from Figure 7a, there is no correlation can be found between acceleration time-history curves (blue color) and the wind speed (orange color) on 5 January 2022, when the highest wind speed was only 2 m/s. Thus, the light wind can pose no effect on the Great wall. 

According to Figure 7b, a high correlation can be discovered between acceleration time-history curve and the mean wind speed curve (greater than 9 m/s). The moment of highest wind speed coincides with the peak time of acceleration time-history curve on 10 January 2022. Hence, it can be concluded that acceleration and high winds are positively correlated. Moreover, the wind with a speed greater than 9 m/s can largely influences the stability of the Great wall.

### 3.3. Analysis of Thunderstorms Impact

Figure 8 shows the acceleration time series of vibration frequencies obtained from 100 Hz accelerometer data for Station 67 under thunder-induced vibration for 24 h on 4 August 2022. The change in the acceleration time series occurred at approximately the same period (62,000 s to 67,000 s, about 17:30–18.30 Beijing time), which corresponds with the occurrence of thunderstorms on the day (see Table 3). 

### 3.4. Analysis of Rain Load Impact

We collected the accelerometer data during the heavy rain on 27 June 2022 to figure out the effect of rainfall on the Great Wall (https://rp5.ru/, accessed on 27 June 2022). As can be seen in Figure 9, it rained heavily in the morning, and the rainfall reached a maximum of 12 mm. In addition, as the acceleration time-history curve is smooth and undisturbed, no correlation existed between acceleration and rainfall. 

On 6 January 2022, extreme weather, accompanied by gusty winds and heavy snow slammed across the Huairou district. To analyze the impact of extreme weather on the Great Wall, we downloaded the data of wind speed, rainfall a, and acceleration. As seen from Figure 10, it rained until 1:00 p.m. on the day, and the maximum rainfall reached 7.5 mm. After the rain, a strong and persistent wind at a speed ranging from 7 m/s to 10 m/s started to blow. It can be seen from Figure 10 that the rainfall curve has no correlation with the acceleration time-history curve, and there is an obvious correlation between the wind speed curve and acceleration time-history curve. The above analysis further verifies that wind is the main factor affecting the stability of the Great Wall, while it is difficult to analyze the impact of rainfall on the Great Wall using high-frequency accelerometer data.

## 4. Conclusions

In this paper, a series of MEMS accelerometers were used to monitor the vibrations of the Great Wall in Huairou District, Beijing. The impacts of traffic vibrations, wind load, thunderstorms, and rain load on the Great Wall were analyzed. Based on the results, traffic vibrations, thunderstorms, and wind load can all affect the stability of the Great Wall. To be more specific, the intensity of the vibrations increases with traffic flow, and enhanced thunderstorm excitation can result in increased vibration displacement. What is more, the effect of wind load gradually increases with the wind level. In addition, there is no correlation between accelerometer data and rainfall data in the short term. 

To sum up, some recommendations can be made to improve the security of the Great Wall. First, anti-vibration or shock-absorbing measures are necessary for the near-road wall. For instance, shock-absorbing rubber speed bumps should be installed on roads near the Great Wall. Also, we should restrict the passage of heavy trucks and strengthen the vibration monitoring of the near-road Great Wall. Then, we need to enhance real-time monitoring and early warning systems of the Great Wall in key protection areas on days with thunderstorms and wind.

## Figures and Tables

**Figure 1 sensors-23-02179-f001:**
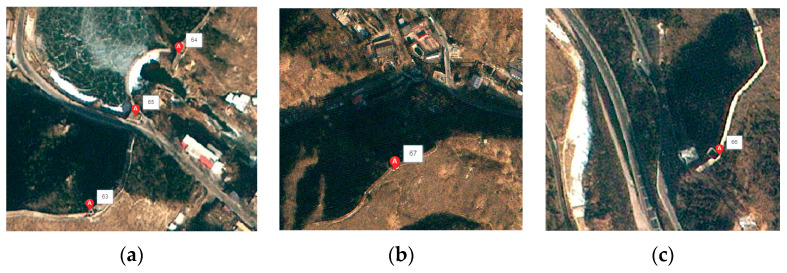
Deployment of the monitoring systems: (**a**) Erdaoguan Great Wall; (**b**) Qiliankou Great Wall; and (**c**) Hefangkou Great Wall.

**Figure 2 sensors-23-02179-f002:**
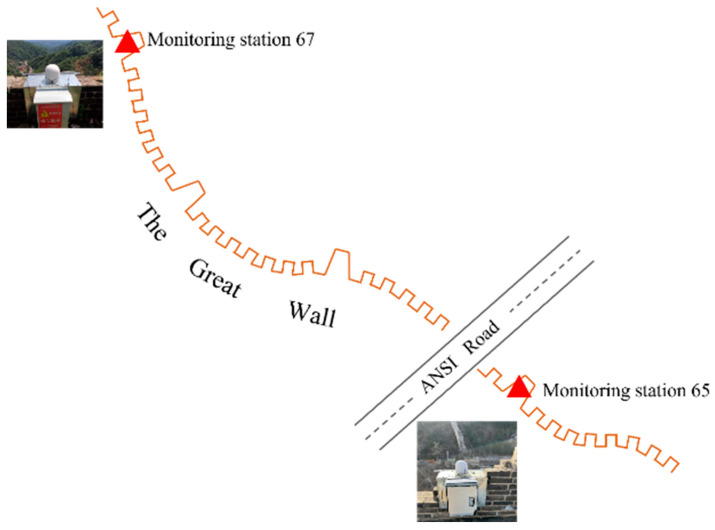
The layout of GNSS and 3-axis Accelerometer.

**Figure 3 sensors-23-02179-f003:**
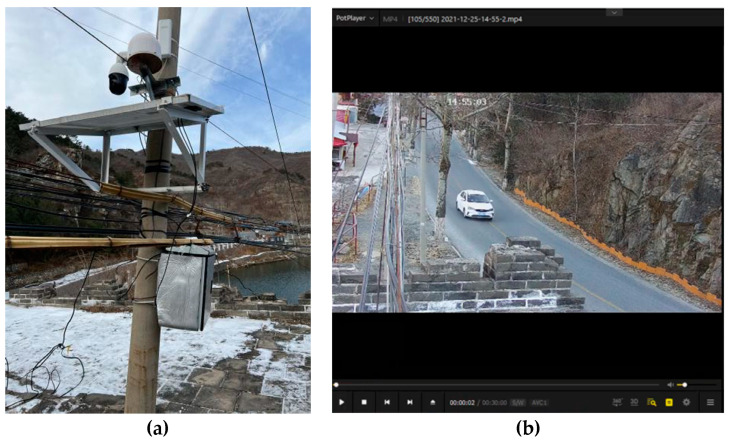
The video camera monitor and video images: (**a**) the video camera monitor; and (**b**) the video images.

**Figure 4 sensors-23-02179-f004:**
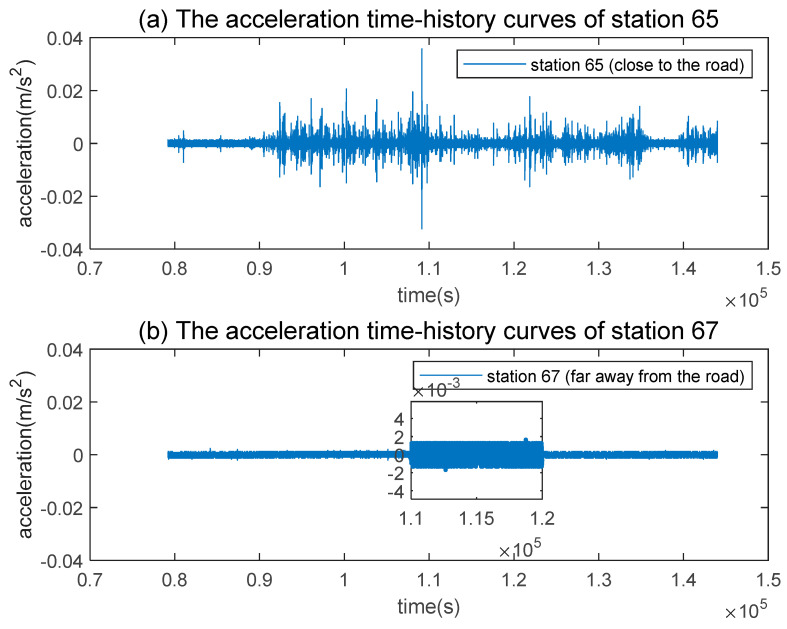
Comparison of acceleration time-history curves between near-road monitoring station 65 (**top**) and far road monitoring station 67 (**bottom**).

**Figure 5 sensors-23-02179-f005:**
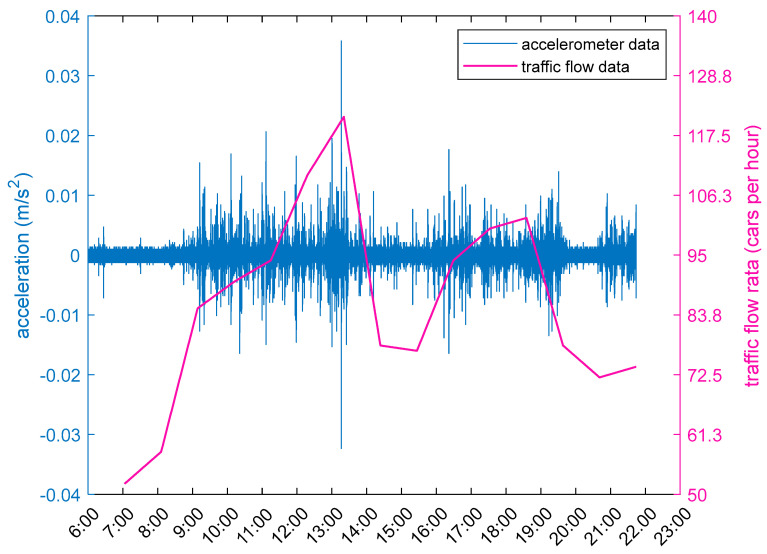
Correlation between acceleration and traffic flow.

**Figure 6 sensors-23-02179-f006:**
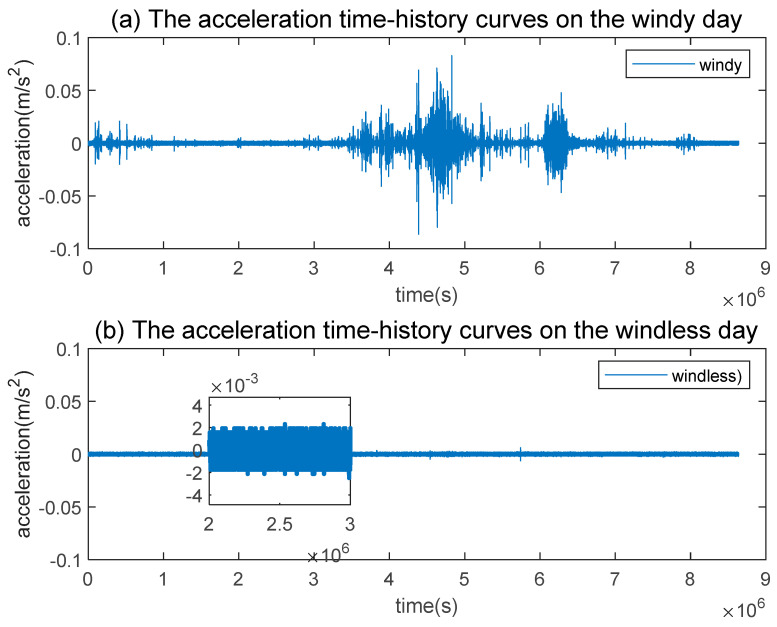
Comparison of acceleration time-history curves on windy and windless day.

**Figure 7 sensors-23-02179-f007:**
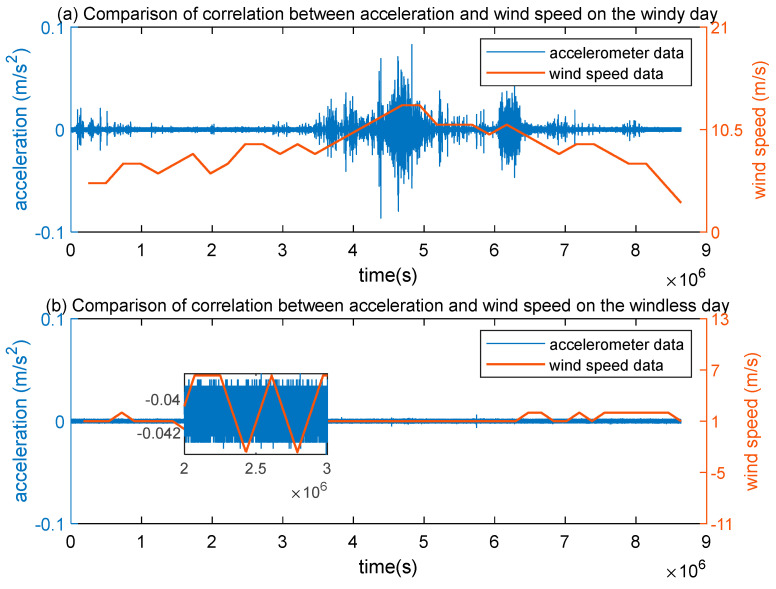
Comparison of the correlation between acceleration and wind speed data on breeze day.

**Figure 8 sensors-23-02179-f008:**
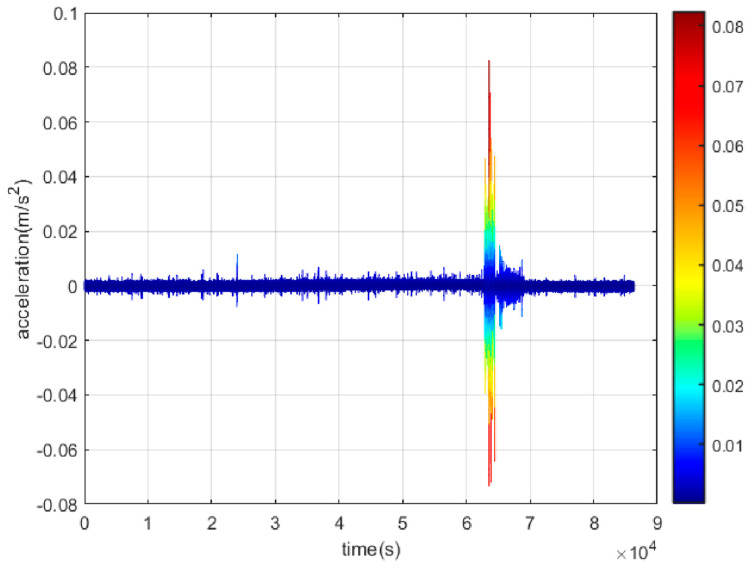
Acceleration time series under thunder loads.

**Figure 9 sensors-23-02179-f009:**
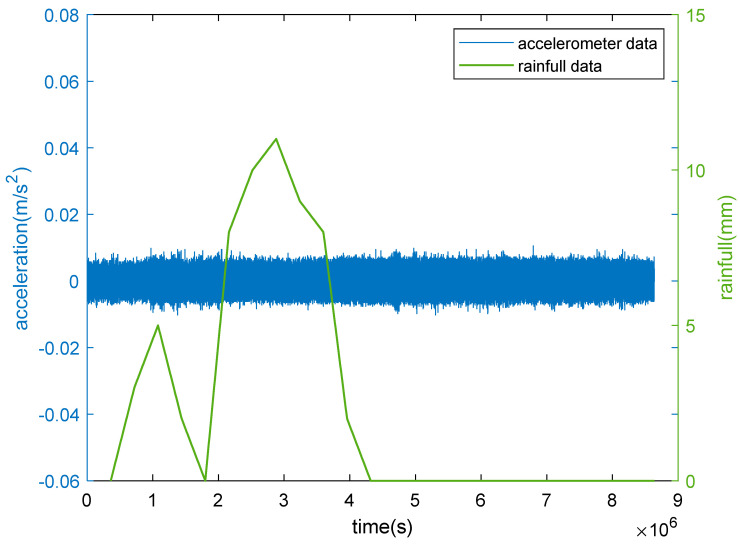
Comparison of the correlation between acceleration and rainfall data.

**Figure 10 sensors-23-02179-f010:**
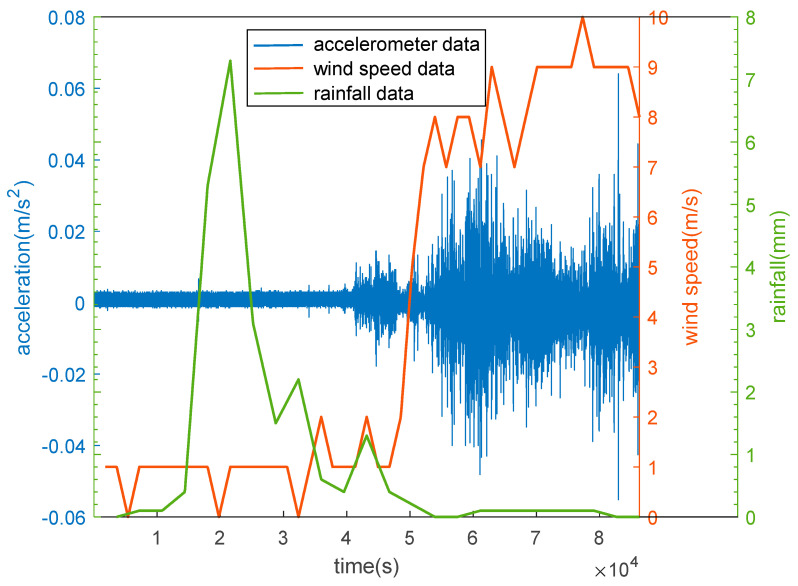
Comparison of the correlation between the acceleration, wind speed, and rainfall data on extreme weather.

**Table 1 sensors-23-02179-t001:** Principal specifications of GNSS and accelerometer.

Parameter	Condition	Value	Unit
Measurement range		±1.5 g, ±3 g, ±6 g	g
Size		7.6 × 8.6 × 3.3 mm	
Offset		0	LSB
Offset error	−40 °C … +125 °C	−20~+20	mg
Sensitivity	±3 g Mode 1 ±6 g Mode 2 ±1.5 g Mode 3	2700 1350 5400	LSB/g
Sensitivity error	−40 °C … +125 °C Mode 1 (±3 g 70 Hz)	−0.7–+0.7	%
Linearity error	−1 g … +1 g range −6 g … +6 g range	−1–+1 −15–+15	mg
Integrated noise		0.44	mg RMS
Noise density		37	$\mu g/\sqrt{\mathrm{Hz}}$
Updating frequency		100	Hz

**Table 2 sensors-23-02179-t002:** Traffic flow on Ansi Road on 31 January 2021.

Beijing Time	Times (s)	Traffic Flow
6:00–7:00	82,801	50
7:00–8:00	86,401	52
8:00–9:00	90,001	58
9:00–10:00	93,601	85
10:00–11:00	97,201	90
11:00–12:00	100,801	94
12:00–13:00	104,401	110
13:00–14:00	108,001	121
14:00–15:00	111,601	78
15:00–16:00	115,201	77
16:00–17:00	118,801	94
17:00–18:00	122,401	100
18:00–19:00	126,001	102
19:00–20:00	129,601	78
20:00–21:00	133,201	72
21:00–22:00	136,801	74

**Table 3 sensors-23-02179-t003:** The weather conditions for 4 August 2022 in Huairou, Beijing.

Beijing Time	4 August 2022		
	Wind Speed (m/s)	Wind Speed (m/s)	Severe Weather
0:00	Light air (1 m/s)	Light breeze (2 m/s)	
0:30	calm	light air (1 m/s)	
1:00	Light air (1 m/s)	light air (1 m/s)	
1:30	Light air (1 m/s)	light breeze (2 m/s)	
2:00	Light breeze (2 m/s)	light air (1 m/s)	
2:30	Light breeze (2 m/s)	light air (1 m/s)	
3:00	Light air (1 m/s)	light air (1 m/s)	
3:30	Light air (1 m/s)	light breeze (2 m/s)	
4:00	Light air (1 m/s)	light breeze (2 m/s)	
4:30	Light air (1 m/s)	light breeze (2 m/s)	
5:00	Light breeze (2 m/s)	light breeze (2 m/s)	
5:30	Light breeze (2 m/s)	light breeze (2 m/s)	(weak) shower, rain
6:00	Light air (1 m/s)	light breeze (3 m/s)	
6:30	Light air (1 m/s)	light breeze (2 m/s)	
7:00	Light air (1 m/s)	light breeze (2 m/s)	
7:30	Light air (1 m/s)	light breeze (2 m/s)	
8:00	Light air (1 m/s)	light breeze (2 m/s)	
8:30	Light air (1 m/s)	light breeze (3 m/s)	
9:00	Light air (1 m/s)	light breeze (3 m/s)	
9:30	Light breeze (2 m/s)	light breeze (2 m/s)	
10:00	Light breeze (2 m/s)	light breeze (2 m/s)	
10:30	gentle breeze (3 m/s)	light breeze (2 m/s)	
11:00	Light breeze (2 m/s)	light breeze (3 m/s)	
11:30	Light breeze (2 m/s)	light breeze (3 m/s)	
12:00	Light breeze (2 m/s)	gentle breeze (4 m/s)	
12:30	Light breeze (2 m/s)	gentle breeze (4 m/s)	
13:00	Light breeze (2 m/s)	gentle breeze (4 m/s)	
13:30	Light breeze (2 m/s)	gentle breeze (4 m/s)	
14:00	Light breeze (2 m/s	gentle breeze (4 m/s)	
14:30	Light breeze (2 m/s)	gentle breeze (4 m/s)	
15:00	gentle breeze (3 m/s)	gentle breeze (4 m/s)	
15:30	gentle breeze (4 m/s)	gentle breeze (4 m/s)	
16:00	gentle breeze (5 m/s)	gentle breeze (4 m/s)	
16:30	moderate breeze (6 m/s)	gentle breeze (4 m/s)	
17:00	gentle breeze (4 m/s	gentle breeze (5 m/s)	
17:30	light breeze (2 m/s)	light breeze (2 m/s)	
18:00	gentle breeze (5 m/s)	light breeze (3 m/s)	
18:30	moderate breeze (7 m/s)	light breeze (2 m/s)	
19:00	moderate breeze (6 m/s)	moderate breeze (7 m/s)	thunderstorm
19:30	gentle breeze (5 m/s)	gentle breeze (5 m/s)	(Weak) thunderstorm, rain
20:00	light breeze (3 m/s)	light breeze (3 m/s)	(Weak) thunderstorm, rain
20:30	light air (1 m/s)	light breeze (3 m/s)	thunderstorm, rain
21:00	light air (1 m/s)	gentle breeze (4 m/s)	(Weak) thunderstorm, gust, rain
21:30	light breeze (3 m/s)	light breeze (3 m/s)	thunderstorm
22:00	light air (1 m/s)	light breeze (2 m/s	
22:30	light breeze (3 m/s)	light breeze (2 m/s)	
23:00	light breeze (3 m/s)	light air (1 m/s)	
23:30	light breeze (3 m/s)	light air (1 m/s)	

## Data Availability

The data presented in this study are available on request from the corresponding author.

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
