# Peer review of "The Great Wall Vibration Monitoring of Traffic and Natural Hazards Using MEMS Accelerometers"

_sensors, 2023, doi:10.3390/s23042179_

Round 1

Reviewer 1 Report

The paper presents an analysis of experimental MEMS acceleration measurements in order to evaluatethe  vibrations. Some correlation with traffic condition and atmospheric meteorological conditions have been provide.

The paper is interesting but some improvement are necessary. In particular It is very difficult for the readers to understand the difference between the two acceleration or coditions because different limits are used for y-axis. In particular

-          Fig 3. Pag. 4. Please use the same y-axis limits for the two figure and add a zoom subfigure for the bottom one. Add a subfigure index (a-b) and caption.

-          Fig 4 . Please use an hour time scale for Fig 4. Moreover  It is not also clear the time in second in table 1.

-          Fig 5. Pag. 6 Please use the same y-axis limits for the two figure and add a zoom subfigure for the bottom one. Add a subfigure index (a-b) and caption.

-          Fig 6 and 7 Please use the same y limits. And for fig 6 if it is necessary to show the variation add a zoom on a part of the graph.

-          Fig 9-10. Use the same y limits of the Fig 6 and 7

As final remarks, it is interesting to summarize the comparison in a final table in terms of percentage acceleration variations in different condition, etc

Reviewer 2 Report

The article examines the correlation between acceleration and traffic vibration, strong winds, heavy rain, thunderstorms based on the acceleration data.

 There are the following concerns, which author need to clarify and improve:

1.      In the introduction, authors state that’ Chen et al. used a seismograph as the main instrument concluded that the  vibration of the Great Wall caused by ground traffic increases with the increasing speed  and weight of the vehicle and decreases with the increasing distance of the vibration source’ . Author should emphasize the advanced effect of using accelerometer in your proposed approach for vehicle vibration concerns. Why should authors utilize 3-axis MEMS accelerometers for this specific case of Great Wall ?

2.      Also, authors need to refer more in detail about other articles, which also used MEMS accelerometer for the vibration measurement. Author should add another section ‘Relative works with deep descriptions about the current work and other related works to enhance paper quality. More references article need to be read.

3.      In the ‘Equipment’ part, authors should create a table, including all the utilized  equipment such as accelerometer,  video camera, GNSS, etc., with their citation references. Also, authors should indicate more precisely the position of the installed accelerometer (Ex: By illustrating of figure,…)

4.      The correlation coefficient ? should be recorded in a summary table for all the examined factors ( traffic and natural factors) with detailed analysis.

5.       What is the frequency of the accelerometer for all tests? Did the authors consider the inducted noise?

6.      What technique calculates the number of car per hour with the video camera?

7.      In the Figure 3, authors must use English to name the X axis.

8.       Authors should put Figure 6 and Figure 7 in the same order of the windy day and windless day like the previous order of Figure 5.

9.      Name of Figure 5 must be ‘ Comparison of acceleration time-history curves for the windy day and windless day’, not ‘ between’.

10.   In the ‘Analysis of Thunderstorms impact’, stations numbered 63, 64, 66, and 67. In section 3 only mentions station 65 and 67. Authors must add the description of station 63,64,66.

11.   In section 4.3, the author state that’ Figure 8 shows a set of vertical acceleration time series of vibration frequencies obtained from 100 Hz accelerometer data for four stations numbered 63, 64, 66, and 67’. What is the relative function between all these stations? How they link together to acquire the acceleration signal from a single accelerometer?

12.   Author state that ‘, The acceleration time series change greatly nearly at 72000s - 75600s’. However, Figure 8 shows the significant change at the period of 62000s to 67000 s. Please verify this information.

13.   The ‘ Weed Speed’ in Table 2 seems not the right term to indicate. Authors can alternate this term.

14.   There are considerable number of grammar issues and phrase descriptions in the paper.

Ex: In line 34 the sentence ‘Although the vibrations induced by traffic are small, the old Great wall, with its accumulated damage, has grown weaker in seismic resistance’ should be rewritten properly to be clearer. In line 157,  it should be ‘there is significant correlation’ not ‘some’, etc.

Authors must double-check the English grammar and format of the manuscript carefully.

Reviewer 3 Report

The authors present a study of detecting the Great Wall by recording vibrations caused by natural weather and traffic with MEMS accelerometers. The author used a low-cost MEMS accelerometer to record the vibration of the Great Wall and made a correlation study between the vibration data and the monitored environmental variables. This is extremely meaningful work for the protection of ancient buildings.

1.

The authors introduce the Spearman correlation analysis method used in this article, but the correlation coefficient is only shown in the wind load impact section. There is no direct data support for the correlation analysis of traffic flow, precipitation, and thunderstorms. It seems that these correlation conclusions were straight summarized from the figures.

2.

The parameters of the MEMS sensor have a great influence on vibration monitoring, and the authors have to show more detailed experimental conditions in the text. For MEMS accelerators, vibrations above the monitored frequency (buildings are more sensitive to low-frequency vibrations) may introduce disturbances and measurement errors. The reasons for this are mainly the conversion of acceleration to velocity magnitude (integral), self-resonant frequency, and improper filter roll-off, which attenuates high-frequency components. Some errors are inherent in the internal structure of MEMS accelerometers and therefore unavoidable.

Is the frequency response range of the SCA3300 considered in the experimental design?

3. 

The authors believe that 'The data of traffic vibration, wind load, and rainfall load impact factors follow a highly volatile random distribution.' (page 2 line 70). Perhaps this statement is correct in the selected data. But normal distribution requires QQ Plot or Kolmogorov Smirnov test verification. The rainfall in the Huairou area occurs less on an annual scale. Generally speaking, the occurrence of rainfall may be more in line with the Poisson distribution.

4. 

The baseline of the accelerometer in the full text is very unstable. For example, in Figure 5, the baseline is about -0.04 m/s2 and in Figure 3 the baseline is 0.02 m/s2. Is it caused by temperature changes? If so, how does temperature affect the sensitivity of the sensor? Due to the different collection times of experimental data such as traffic and wind speed, the measured acceleration will lose comparability due to changes in the test environment temperature.

5. 

On page 7 line 165, where are station numbers 63, 64, and 66 located? It is not marked in Figure 1.

6. 

The thunderstorm intensity in Figure 8 lacks a scale bar.

7. 

How did the authors conclude that traffic is the biggest influencing factor? Is it according to the maximum value of the acceleration intensity or the integral area? If it is based on the integral area, the data of only one day is not enough to explain the impact of the whole year. Similarly, if it is based on intensity, it is difficult to prove that other conditions (extreme weather) cannot produce a greater acceleration value.

Round 2

Reviewer 2 Report

The authors already put the high effort to improve the manuscript based on the previous comments. There are some additional comments as follow:

1.      The tables must follow the template of MDPI sensors especially table 1 and 2. I suggest authors to use the table from the journal template.

2.      Please add x-axis label to the Figure 4 b

3.      Authors removed the description part of correlation coefficient ?, but still mention it in other parts such as

In line 157 ‘correlation coefficient reaches a value of 0.94’

Etc. Author must check this info in other parts also.

Furthermore, since no explanation of correlation coefficient ?, the correlation between acceleration and other events are mainly­ based on the chart, which are from the collected data. Thus, authors must emphasize your approach method in the abstract and introduction to convince the reader. You can add more references to strengthen the paper quality.

4.      After all the content modification, authors must double-check the English grammar and format of the manuscript one more time carefully.

Reviewer 3 Report

After the revision, the quality of the article has been greatly improved. All my comments have been clearly addressed.
